# Efficient Ocular Delivery of VCP siRNA via Reverse Magnetofection in RHO P23H Rodent Retina Explants

**DOI:** 10.3390/pharmaceutics13020225

**Published:** 2021-02-06

**Authors:** Merve Sen, Marco Bassetto, Florent Poulhes, Olivier Zelphati, Marius Ueffing, Blanca Arango-Gonzalez

**Affiliations:** 1Centre of Ophthalmology, Institute for Ophthalmic Research, University of Tübingen, 72076 Tübingen, Germany; merve.sen@uni-tuebingen.de; 2Graduate Training Centre of Neuroscience, University of Tübingen, 72074 Tübingen, Germany; 3OZ Biosciences, Parc Scientifique de Luminy, CEDEX 9, 13288 Marseille, France; mbassetto@ozbiosciences.com (M.B.); fpoulhes@ozbiosciences.com (F.P.); ozelphati@ozbiosciences.com (O.Z.)

**Keywords:** siRNA delivery, magnetic nanoparticles, magnetofection, RNAi therapy, ocular therapy, retinal organotypic culture, retinal degeneration, retinitis pigmentosa, RHO P23H

## Abstract

The use of synthetic RNA for research purposes as well as RNA-based therapy and vaccination has gained increasing importance. Given the anatomical seclusion of the eye, small interfering RNA (siRNA)-induced gene silencing bears great potential for targeted reduction of pathological gene expression that may allow rational treatment of chronic eye diseases in the future. However, there is yet an unmet need for techniques providing safe and efficient siRNA delivery to the retina. We used magnetic nanoparticles (MNPs) and magnetic force (Reverse Magnetofection) to deliver siRNA/MNP complexes into retinal explant tissue, targeting valosin-containing protein (VCP) previously established as a potential therapeutic target for autosomal dominant retinitis pigmentosa (adRP). Safe and efficient delivery of VCP siRNA was achieved into all retinal cell layers of retinal explants from the RHO P23H rat, a rodent model for adRP. No toxicity or microglial activation was observed. *VCP* silencing led to a significant decrease of retinal degeneration. Reverse Magnetofection thus offers an effective method to deliver siRNA into retinal tissue. Used in combination with retinal organotypic explants, it can provide an efficient and reliable preclinical test platform of RNA-based therapy approaches for ocular diseases.

## 1. Introduction

Advances in understanding RNA interference (RNAi)-based signaling pathways have opened up new therapeutic perspectives for a wide range of ocular disorders. The use of small interfering RNA (siRNA)-induced gene silencing has great potential in the treatment of chronic eye diseases such as some forms of retinitis pigmentosa (RP), age-related macular degeneration (AMD), diabetic retinopathy and glaucoma, which are associated with aberrant activities of specific proteins [1,2,3]. The eye is indeed an attractive target organ for RNA-based therapies. As the area to be treated is usually small and the success of the treatment can be easily controlled, it allows local treatment with smaller doses. Besides, a relative immune privilege due to the blood–retina barrier and low lymph drainage minimizes the risk of undesirable side effects [4]. However, despite these properties, the delivery of siRNA to the posterior segment of the eye, especially to the retina, has proven to be challenging. Given the low bioavailability of siRNAs due to their high susceptibility to enzymatic hydrolysis, poor cellular uptake and rapid elimination from the circulatory system, as well as potential undesirable side effects due to off-targeting and immunogenicity, siRNA delivery to the retina has encountered many hurdles [5]. In fact, sufficient siRNA bioavailability in this compartment cannot be achieved by either topical [6] or systemic administration [7]. Instead, the direct application of siRNA into the posterior chamber by intravitreal injection (IVT) is constrained by the short half-life (2 days) of siRNAs in the vitreous [7]. When siRNA elimination and intraocular metabolism are taken into account, only a very small fraction of the siRNA dose is available in the retina [8]. Therefore, repeated injections are unavoidable but impose a significant burden on patients’ compliance and increase the potential risk of cataracts, retinal detachment, and other eye injuries [9]. To a certain extent, these shortcomings can be compensated by integrating the siRNA into a carrier, which leads to the formation of siRNA/carrier complexes. However, it has to be considered that the compatibility of a particular carrier depends on the tissue of interest. Therefore, careful selection of the physiologically relevant tissue model proves necessary to study the distribution and effects of siRNA. Organotypic retinal explant cultures are a very reliable platform to screen and test new therapeutic agents, combining the advantages of in vitro and in vivo models by providing structural and functional features and properties of the entire retina [10]. Retinal organotypic cultures sustain the three-dimensional architecture and cellular connections within the tissue for up to 3 weeks in vitro [11] and significantly diminish the number of animals per experiment. Thus, this system serves as a well-defined experimental platform to test the suitability of a particular carrier for siRNA delivery.

However, siRNA delivery to retinal explants is not easy to achieve. Nonviral carriers have the advantages of low immunogenicity, easy production and low costs, but do not reach all cell layers equally [12]. When applied to the medium from the posterior retina pigment epithelium (RPE) side or directly at the ganglion cell layer (GCL) at the anterior side of the retina, nonviral carriers only reach either the photoreceptors (PRs) [13] or the ganglion cells [14], respectively. Physical transfection methods, such as gene-gun or electroporation, fail to preserve cell membrane integrity [15]. Virus-based vectors are presently the most encouraging option for siRNA delivery in treating retinal disease [16,17]. Nonetheless, viral vectors require invasive subretinal injections that include topical retinal detachment, and their potential risk regarding immunogenicity, readministration, large-scale production, and high manufacturing costs make these methods cumbersome [2]. Consequently, there is a clear need for an effective, safe, and noninvasive method of administering siRNAs that can ascertain its effective delivery into deeper retinal layers. Magnetic force driven transfection commercialized as Magnetofection™ [18] combines the advantages of biochemical, lipid or polymer-based transfection and physical transfection methods while excluding some of their inconveniences (low efficiency, toxicity). Based on the insertion of nucleic acids into magnetic nanoparticles (MNPs) and applying physical magnetic force, it can concentrate and transport siRNA into deeper retinal layers supported by an appropriate magnetic field [19]. Autosomal dominant retinitis pigmentosa (adRP), which is associated with abnormal activities of mutated proteins [20], could specifically benefit from siRNA-mediated gene silencing. Retinitis pigmentosa (RP) is a group of hereditary eye diseases characterized by gradual loss of photoreceptor neurons and blindness [21]. Previously, in an RP model of *Drosophila*, in which the mutant *Rh1^P37H^* corresponds to the P23H mutation in the human rhodopsin (*RHO*) gene, we were able to show that excessive activity of valosin-containing protein (VCP) could be a pathological driver of photoreceptor cell death. In this context, we could verify that reduced *VCP* levels in these flies confer neuroprotection [22]. Our current work proves that pharmacological inhibition of VCP can also delay retinal degeneration in the RHO P23H rat in vivo [23]. Using an RNAi based approach, we now provide evidence, that silencing *VCP* expression in RHO P23H transgenic rat organotypic retinal cultures via Reverse Magnetofection can protect photoreceptor cell death and attenuate retinal degeneration in vitro. OZ Biosciences recently developed MNPs, called XPMag, which have been optimized for safe and efficient siRNA delivery to retinal cell lines by Magnetofection [24] (Magnetofection is a trademark of OZ Biosciences). This method features high silencing efficiency for *GFP* and *GAPDH* and low toxicity in retinal cell lines. Here we explored gene silencing of *VCP* via Magnetofection in RHO P23H rat retinal explants. To achieve this, we first tested MNPs to assess their efficacy in two retinal cell lines, the mouse immortalized cone photoreceptor cell line 661W and the human immortalized retinal pigment epithelial cell line hTERT-RPE1. Subsequently, we established delivery of VCP siRNA/XPMag complexes through the posterior, RPE side of retinal explants.

## 2. Results

### 2.1. Characterization of VCP siRNA/XPMag Complexes

We recently generated MNPs for safe and efficient siRNA delivery to cells and tissues assisted by magnetic targeting. In a first step, we could show that our MNPs termed XPMag could form stable complexes with siRNA and efficiently silenced the reporter gene *GFP* and the housekeeping gene *GAPDH* in retinal cell lines [24]. Here, we now aim to test this new method to silence the *VCP* gene, a potential target gene for therapeutic intervention in RP. MNPs assemble nucleic acids through noncovalent hydrophobic and electrostatic interactions, resulting in the formation of complexes with a hydrodynamic size in the nanometer range and a net electrical charge in the millivolt range, which are compatible with cellular uptake. The complex formation of MNPs with siRNA depends on the specific siRNA as well as the siRNA concentration used. To characterize complexation of VCP siRNA with XPMag nanoparticles, we first measured the hydrodynamic size and the surface electrostatic charge of VCP siRNA/XPMag complexes formed with 25 nM VCP siRNA after 30 min of complexation since we had previously obtained good results for this concentration with GFP siRNA [24]. As shown in Figure 1, the hydrodynamic size of the XPMag particles, measured by dynamic light scattering (DLS), did not change by complex formation with 25 nM VCP siRNA (Figure 1A), while the surface electrostatic charge shifted towards a lower zeta potential without alteration of the peak shape (Figure 1B). The modification in XPMag surface electrostatic charge demonstrates the complexation of siRNA to the MNPs since the negatively charged nucleic acids partially mask the positively charged surface of the MNPs. As observed previously with GFP siRNA, also with VCP siRNA, the zeta potential remained in high positive values, and particles between 40 and 60 mV are known to show good stability behavior [25], which indicates that at a concentration of 25 nM siRNA, the stability of the complexes is not affected. When analyzing the complex formation with increasing VCP siRNA concentrations, we found that complexes with similar hydrodynamic sizes were formed with VCP siRNA concentrations of 5–100 nM (Figure 1C). On the other hand, the surface electrostatic charge slightly decreased in a siRNA dose-dependent manner, as measured by zeta potential values starting from 58 mV to reach 48 mV at 100 nM VCP siRNA (Figure 1D), but remained in a highly positive range with all VCP siRNA concentrations tested. These results confirm our previous results with GFP siRNA and show that stable VCP siRNA/MNP complexes are formed at siRNA concentrations up to 100 nM. The morphological aspect of XPMag was characterized by TEM. No significant differences were observed with or without siRNA complexation (data not shown). XPMag formulation is composed of spherical particles with an average diameter of 180 nm (Appendix A). The size measured by this technique was slightly smaller compared to the hydrodynamic diameter measured by DLS but remained in the same range.

### 2.2. Classic Magnetofection to Silence VCP in Retinal Cell Lines

In the next step, we tested the applicability of classic Magnetofection method to silence *VCP* in vitro in two different retinal cell lines. We used XPMag complexes formed with 25 nM VCP siRNA and the classic Magnetofection protocol, where transfection is achieved by a magnet placed under the cell culture plate to attract the MNPs/siRNA complexes from the culture medium into the cells (Figure 2A,B and Appendix A). Here we compared the efficiency of Magnetofection of VCP siRNA/XPMag in *VCP* silencing with three other different groups: (1) an untreated group, to determine the baseline level of cell viability and phenotype; (2) a lipid-based transfection group, VCP siRNA was complexed with Lullaby—a lipid-based siRNA transfection reagent—to compare the silencing efficiency of lipofection (lipid-based transfection) with Magnetofection; (3) a negative control group, here, scrambled siRNA—randomized nucleotides not targeting known genes—was complexed with either XPMag (by classic Magnetofection) or Lullaby.

We monitored the transfection efficiency of VCP siRNA/XPMag by fluorescence-microscopy using VCP immunostaining 72 and 48 h post-transfection, both in the mouse photoreceptor cell line 661W (Figure 2C) and in the human retinal pigment epithelium cell line hTERT-RPE1 (Figure 2D), respectively. The results initially ensured that the XPMag transfection did not induce toxicity because the cell-monolayer integrity remained intact, as observed in bright-field microscopy imaging.

In parallel, fluorescent microscopy imaging for VCP staining confirmed the efficiency and specificity of the silencing effect. Compared to untreated cells, cells transfected with XPMag alone or with scrambled siRNA showed no change in VCP fluorescence intensity in both mouse retinal 661 W cells (Figure 2C) human hTERT-RPE1 cells (Figure 2D). In contrast, VCP staining showed a substantial decrease in VCP siRNA/XPMag conditions compared to untreated or RNAi-scrambled transfected controls. Compared to transfection with the lipofection reagent Lullaby, classic Magnetofection achieved an evidently higher *VCP*-silencing efficiency in the human hTERT-RPE1 cell line, as well as a comparable efficiency in the mouse retinal 661W cell line (Figure 2C,D). Subsequently, the efficiency of VCP siRNA/XPMag complexes in knocking down *VCP* expression in human hTERT-RPE1 cells was assessed by Western blot analysis. Whole-cell lysates of human hTERT-RPE1 cells transfected with VCP siRNA/XPMag showed a marked decreased VCP expression at both 25 nM (n = 3, data not shown) and 50 nM compared to the untreated or scrambled siRNA control conditions (Appendix A). However, doubling the siRNA dose up to 50 nM significantly further diminished VCP expression compared to the untreated or corresponding scrambled siRNA/XPMag negative control. Quantification and normalization of the Western blotting signal for VCP to the beta-actin intensity showed significant *VCP* silencing through Magnetofection in human hTERT-RPE1 cells (Appendix A). Overall, these results show that the newly developed method using XPMag efficiently silences *VCP* by classical Magnetofection in cultures of a single-layer of retinal cells.

### 2.3. VCP Silencing in Retina Organotypic Cultures of RHO P23H Transgenic Rats by Reverse Magnetofection

Next, we adapted our transfection method to transfect the multilayered cell tissue of retinal organotypic cultures. For this purpose, we used retinal explants from RHO P23H transgenic rats, a model for adRP. The aim was to silence *VCP* as a potential therapeutic target for adRP [22,23]. The RHO P23H transgenic rat model carries the human *RHO* gene with the P23H mutation, leading to rapid and progressive photoreceptor degeneration starting at postnatal (PN) day 9, with cell death peaking at PN15 in heterozygous retinae [26,27]. We have developed ex vivo organotypic retinal cultures from this rat model to investigate pathological mechanisms and test potential therapeutics. The steps of preparation are depicted in Figure 3A. Briefly, after enucleation, only the neural retina and its adherent RPE were transferred to the upper compartment of a Transwell^®^ membrane with the GCL layer upwards and the photoreceptors/RPE side facing the membrane inserts. In a previous study, we could show that partial genetic inactivation of *VCP* in the *Drosophila* RP model *Rh1*^P37H^—corresponding to the P23H mutation in humans—was neuroprotective [22]. Here we aimed to silence *VCP* in the RHO P23H transgenic rat model using organotypic retinal cultures. The thickness of the retina, which is approximately 250 µm [28] in this animal model at PN15, makes ex vivo transfection of retina explants challenging compared to transfection of cell monolayers. Moreover, transfection of retinal tissue by nonviral vectors without a directing force applied either in the culture media (RPE/photoreceptors side) [13] or directly onto the explants (GCL side) [14] can only transfect the RPE/photoreceptors or the ganglion cells, respectively, which means they cannot reach cell layers further away. On the other hand, Magnetofection allows the delivery of nucleic acids either alone or in combination with vectors associated with MNPs under the influence of an external magnetic field [19,29,30]. On this basis, we hypothesized that the transfection of deeper retinal layers in explants, that are not accessible from the surface, could be improved by Magnetofection. To silence *VCP* in retinal organotypic cultures of RHO P23H rats, we approached the retina from the RPE side by adding the siRNA/MNPs complexes to the culture medium and concentrating and drawing them to the retina using Reverse Magnetofection, where a magnet is placed above the cell culture plate. Retinal explants were cultured at PN12 and silenced for 3 days (72 h) until the peak of degeneration at PN15 (Figure 3B). We chose this treatment duration based on efficient *VCP* gene silencing in retinal cell lines within 48–72 h (Figure 2C,D). Retinal explants were prepared and transferred to a porous polycarbonate membrane (RPE layer facing down) (Figure 3A). 

Simultaneously, during the preparation of the VCP siRNA complexes with XPMag (Figure 2A), RHO P23H retinae were allowed to adapt to the medium conditions for 30 min. The complexes were then transferred to the wells and mixed with the medium. Finally, the super-magnetic plate was placed on top of the lid of the culture plate during 30 min (Appendix A). The magnetic force applied from above ensures that the complexes (in the medium) are pulled through the tissue layers from the RPE side up to the GCL side of the explant (Figure 3C). This step aims to deliver siRNA into the deeper retinal layers that are otherwise inaccessible from culture media. The entire transfection was performed in only 1 h and is very easy to handle. The next day, the medium was changed, and explants were incubated for another 48 h before evaluating the transfection effect. In the experiment, retinae were collected for five different groups. One group was treated with Reverse Magnetofection using VCP siRNA complexed with XPMag. In this group, VCP siRNA/XPMag was complexed with 50 nM VCP siRNA since we had previously achieved good results at this concentration with VCP siRNA in human hTERT-RPE1 retinal cell lines by classic Magnetofection. Three groups were used as controls: one received only the medium and the exposure to the magnetic field (not-treated), one was treated with Reverse Magnetofection using 50 nM scrambled siRNA complexed with XPMag (negative control), and one was treated with Reverse Magnetofection but with XPMag alone (not complexed). The last group consisted of retinae treated with lipid-based-Lullaby complexed with 50 nM VCP siRNA to compare the efficiency of Reverse Magnetofection with lipofection in organotypic retinal cultures. The success of *VCP* silencing across the whole retina thickness was assessed by immunostaining of VCP. To standardize the analysis, we restricted the evaluation to the central area of the explants. In the RHO P23H animal model, the central retina displays the photoreceptors’ highest density and is also where the degeneration process begins [31]. As expected, since VCP is a ubiquitously expressed multifunctional protein [32], in the not-treated RHO P23H retinae, VCP immunostaining was observed throughout the retina, being more intense in the GCL and inner nuclear layer (INL) as in the outer nuclear layer (ONL) (Figure 4A). To measure VCP intensity, we quantified the mean peak of fluorescent intensity in the ONL, INL and GCL areas in images taken under identical conditions and calculated the fluorescence intensity and standard deviation. Using Reverse Magnetofection of VCP siRNA/XPMag complexes, the mean peak of fluorescence intensity of VCP decreased significantly in the ONL, INL, but also in the GCL (Figure 4B–D). In the group of RHO P23H retinae treated with Reverse Magnetofection of VCP siRNA/XPMag complexes, the integration of the fluorescence signal resulted in 63%, 70%, and 60% silencing of *VCP* in the ONL, INL, and GCL, respectively, compared to the untreated group. This effect was a direct response to Reverse Magnetofection of VCP siRNA/XPMag complexes, as retinae treated with scrambled siRNA/XPMag complexes or XPMag alone showed similar VCP fluorescence intensity profiles as the untreated group. In addition, the absence of *VCP* silencing in the lipofection-transfected retinae highlighted the critical role of the magnetic force in concentrating, attracting, and orientating the siRNA/XPMag complexes to the explant.

These results show that Reverse Magnetofection using XPMag MNPs can accomplish efficient gene silencing of *VCP* in the whole rat retinal organotypic culture. Importantly, a magnetic force is required to deliver siRNA into the deeper layers of retinae (up to the GCL), that are not in direct contact with the culture medium.

### 2.4. Reverse Magnetofection Is Well Tolerated, and VCP Silencing Shows a Neuroprotective Effect in RHO P23H Organotypic Cultures

To further investigate possible side effects of the Reverse Magnetofection treatment and a potential neuroprotective effect of *VCP* silencing, we evaluated apoptotic cell death in the different treatment groups. To this end, we used terminal deoxynucleotidyl transferase dUTP nick end labeling (TUNEL) staining to identify photoreceptors subject to cell death, which is indicated by green fluorescence. We then determined the percentage of TUNEL positive dying cells compared to the total number of ONL cell nuclei. In the untreated RHO P23H retinae, we found approximately 4% ± 1.91 of TUNEL-positive cells in the ONL, reflecting the RP-related degeneration in the animal model itself and the stress induced by the isolation procedure. Reverse Magnetofection with XPMag particles alone or complexed with scrambled siRNA did not increase cell death compared to the untreated control group, indicating that the transfection protocol is safe and well-tolerated in retinal organotypic cultures (Figure 5A,B). Remarkably, Magnetofection-mediated *VCP* silencing was neuroprotective in this model, shown by a lower number of TUNEL-positive cells compared to the controls (Figure 5B). 

Nevertheless, this effect was not achieved by transfection with the lipofection reagent Lullaby. The reduction in cell death in RHO P23H retinae treated with VCP siRNA/XPMag was also mirrored in an increased number of remaining photoreceptor cell rows in the ONL. Therefore, we quantified the cell rows by counting DAPI-stained cells in a linear series in vertical sections of RHO P23H explants. In this way, we found that the ONL of VCP siRNA-treated RHO P23H retinae by Reverse Magnetofection contained significantly more cell nuclei rows than the other groups. This emphasized the neuroprotective effect of *VCP* silencing on photoreceptor cell survival in the RHO P23H animal model, as we confirmed by the reduced cell death. A nonsignificant difference in the number of photoreceptor cell rows in the Reverse Magnetofection of XPMag alone or scrambled siRNA/XPMag compared to the nontreated group (Figure 5C) confirmed the lack of cell mortality upon treatment. Compared to transfection with the lipofection reagent Lullaby, Reverse Magnetofection of VCP siRNA achieved significantly higher photoreceptor cell growth in RHO P23H retinal cultures. Since Reverse Magnetofection is conducted with RPE to GCL directionality, and *VCP* silencing is statistically significant in all retinal explant layers (Figure 4), we also checked cellular toxicity in different layers of the retina after the application of Reverse Magnetofection. For this purpose, we calculated the percentage of cell death by TUNEL assay and photoreceptor cell growth in INL and GCL. Reverse Magnetofection illustrated the absence of cellular toxicity in INL and GCL compared to the untreated control group (Appendix A). In addition, the neuroprotective effect of *VCP* silencing is also found significant in the INL in the retinae treated with VCP siRNA/XPMag. These results demonstrate that Reverse Magnetofection is an efficient, robust, and nontoxic strategy for *VCP* silencing in different retina layers, which is superior to lipofection. In the organotypic RHO P23H model for adRP, the efficient silencing of *VCP* showed a neuroprotective effect with improved photoreceptor cell survival in the ONL and INL.

### 2.5. Effects of Reverse Magnetofection on Astrocytes, Müller Cells, and Microglial Cells in RHO P23H Retina Organotypic Cultures

For the continued evaluation of the application of Reverse Magnetofection and the effects of *VCP* silencing in RHO P23H retinae, we investigated possible retinal stress and inflammatory effects. To this end, we performed immunofluorescence staining for glial fibrillar acidic protein (GFAP) and ionized calcium-binding adapter molecule 1 (Iba1) to evaluate astrocytes, Müller cells, and microglia in retinal tissue. In response to infection and in certain inflammatory scenarios, Müller cells and retinal astrocytes undergo reactive gliosis, being GFAP expression in those cells a typical feature of retinal stress [33,34]. Microglia fulfills a key role in controlling immune responses in the retina, and Iba1 is a specific marker for microglial activation which is associated with a local proinflammatory environment [33,35,36]. To assess glial activation following Reverse Magnetofection-based *VCP* silencing, we labeled treated and not treated RHO P23H retinae using specific antibodies against GFAP. In the RHO P23H transgenic rat, the number of astrocytes is higher than in the age-matched wild type (WT) control, which supports the concept of astrocyte proliferation [35]. The GFAP immunoreactivity propagating from the GCL through the ONL indicates activation of Müller cells and retinal gliosis [34], as seen in the untreated RHO P23H explants (Figure 6). Similar patterns were observed in the RHO P23H retinae treated with XPMag alone, indicating that MNPs did not activate retinal gliosis in retinal explants. To measure activation of retinal gliosis, we quantified the mean peak of fluorescent intensity of GFAP in the GCL and ONL. Retinae treated with VCP siRNA/XPMag or XPMag alone showed similar values indicating that MNPs and Magnetofection with or without downregulation of *VCP* did not activate retinal gliosis in retinal explants (Figure 6B). However, we found increased gliosis in retinae treated with scrambled siRNA/XPMag. Similarly, microglia were assessed in situ by staining retinal cross-sections using specific antibodies against Iba1. At rest, microglial cells exist in the multiple-branched form localized only in the inner/outer plexiform layers (IPL and OPL) and the GCL of the retina [36]. However, when activated, microglial cells become big round blobs which are recruited and migrate through the ONL to promote inflammation [33,37]. To understand microglial activation after *VCP* silencing through Reverse Magnetofection, we considered two parameters: microglial cells with ameboid morphology (round-shaped) and/or displacement to the ONL. As can be seen from the GFAP pattern, microglial cells were also elevated in RHO P23H rats compared to WT rats and during culture preparation [38]. Iba1 immunostaining in the untreated, scrambled siRNA/XPMag, and XPMag alone groups showed similar microglial activation (Figure 6C). *VCP* silencing via Reverse Magnetofection abated Iba1 activation, with significantly fewer round-shaped Iba-1-positive cells in the GCL and ONL. Consequently, Reverse Magnetofection does not activate retinal gliosis or microglial cell activation, but interestingly, *VCP* silencing through Reverse Magnetofection reduced microglial cell activation and decreased potential inflammation.

### 2.6. VCP Silencing through Reverse Magnetofection Restores RHO Intracellular Distribution

As an additional proof of the functional efficiency of Reverse Magnetofection, we analyzed the effect of *VCP* silencing on RHO expression and localization by immunofluorescence staining. Some of the main features of RHO P23H photoreceptor degeneration are mislocalization of RHO and shortening of the photoreceptor’s outer segment (OS) [39]. Therefore, we analyzed the effect of *VCP* downregulation on RHO expression by staining cryo-sections of untreated, VCP siRNA/XPMag, and scrambled siRNA/XPMag administered RHO P23H transgenic rat retina explants with an anti-RHO antibody. Distribution of RHO immunostaining, mean peak of fluorescence intensity in the ONL, and length of the OS were evaluated. Age-matched WT retinal explants were used as a positive control. Here and as described previously, WT RHO shows a high level of RHO expression localized explicitly at the OS [23] (Figure 7A). Although RHO P23H rat retinae exhibit low RHO expression in the OS, it is instead abnormally accumulated in ONL of photoreceptor cells (Figure 7A). 

To test whether *VCP* silencing can correct the decrease of RHO in OS, we evaluated RHO immunostaining of RHO P23H retinae transfected with VCP siRNA and its scrambled siRNA control. The RHO staining in scrambled siRNA treated control RHO P23H retinae was mislocalized, similar to the untreated retinae. In contrast, *VCP* silencing in RHO P23H retinae almost completely restored RHO’s distribution to that of the WT phenotype, with staining predominantly in the OS. This phenotype is in parallel with the findings in our recent report, which showed that pharmacological inhibition of VCP restores RHO localization [23]. To measure distribution of RHO, we quantified the mean peak of fluorescent intensity in the ONL. To do so, the ONL area was selected in images and the mean fluorescent intensity was calculated. The quantification of RHO immunofluorescence intensity in ONL revealed a significant decrease in the RHO P23H rat retinae treated with VCP siRNA compared to untreated and scrambled siRNA/XPMag treated groups, confirming the reduced retention of RHO in the ONL (Figure 7B). The reduction of RHO distribution in the ONL correlated with an increase of the OS length. The measurements of OS length of VCP siRNA treated RHO P23H retinae displayed showed a longer OS length after *VCP* silencing (Figure 7C). These results demonstrated that the administration of VCP siRNA by Reverse Magnetofection in RHO P23H rat retinal organotypic cultures substantially restored the localization of RHO to the WT state. Together with the results described above, this shows that all major hallmarks essential for reversal of the adRP phenotype can be achieved by *VCP* silencing through Reverse Magnetofection of the VCP siRNA in this organotypic in vitro model of adRP.

## 3. Discussion

Gene silencing by small interfering RNA (siRNA) is one of the most promising techniques to translate the fundamental mechanism of RNAi to clinical applications. RNAi-based therapies using siRNAs to silence aberrant genes are currently being developed for various human diseases, including ocular disorders [2,3]. To date, there are several naked or chemically modified siRNA drugs for the treatment of eye diseases completing Phase II clinical trials, e.g., Tivanisiran for dry eye [3], AGN211745 and PF-045233655 for AMD and QPI-1007 and Bamosiran for glaucoma and ocular hypertension [2], by targeting and silencing different aberrant proteins. Despite their potential in treating ocular diseases, the efficient delivery of siRNAs remains the biggest challenge, especially for retinal delivery. Viral vectors have the great advantage of delivering siRNA across all retinal layers [16,17], but safety concerns and manufacturing costs limit their use [12]. Physical methods, such as electroporation, transfer the genetic material in all layers of cultured retinas when vectors are electroporated from the scleral side of the explant. However, the transfection efficiency is null or very limited from the vitreal side and/or when retinal progenitor cells are fully differentiated into mature retinal cells [40]. Nonviral vectors are safe and easy to mass-produce; however, they are less efficient in gene delivery to the retina as they have limited diffusion into the tissues [12]. Magnetofection makes use of nonviral vectors and physical force by complexing nucleic acids in MNPs and transferring the genetic material by magnetic force acting on these particles [18,19]. Numerous studies have shown the benefits of Magnetofection in vivo, including anticancer immune therapy in cats [41]; treatment of rectoanal motility disorders in rats [42]; treatment of hind limb ischemia in mice [43]. Additionally, in the central nervous system, Neuromag^®^ efficiently delivered functional miRNA inhibitors in rat brain regions surrounding lateral ventricles [44]. In the eye, studies using MNPs for gene delivery in vivo, without any magnetic guidance, have demonstrated that after intravitreal injection, they can successfully be addressed to the choroid in zebrafish [45] or to other ocular tissues, including the retina, in rabbits [46]. Interestingly, applying a magnetic field has been shown to allow MNPs localization in the retina after systemic administration of MNPs in mice [47] and to guide movement towards the cornea after injection of MNPs into the aqueous humor [48]. Besides, controlled delivery of MNP-containing stem cells to the rat retina has been obtained after intravitreal or intravenous injection, by placing a magnet in the upper hemisphere of the ocular orbit [49]. These studies indicated that MNPs can spread within the ocular layers and are well-tolerated without compromising homeostasis or eye functions. However, no approach for the noninvasive administration of MNPs to the retina has yet been developed that could avoid the possible side effects of intravitreal injections. In this study, we demonstrated the feasibility of Reverse Magnetofection as a noninvasive method of gene delivery to the retina in organotypic retinal cultures. We efficiently silenced *VCP* in RHO P23H rat retinal explants by magnetic targeting of a new MNP formulation complexed with siRNA of VCP. We could show that the application of magnetic force from above the culture plate can draw and concentrate the MNP/siRNA complexes from the culture media towards the retinal explant. In this way, all deeper retinal layers (from RPE to GCL) could be efficiently transfected, including those not accessible from the surface. As we did not observe any adverse effects, this transfection method can be further developed for safe in vivo siRNA delivery. It is expected that the diffusion of the siRNA alone or complexed with transfection reagents from the culture media into the explant is not sufficient to achieve gene silencing in all retinal layers. Our study shows the importance of applying a physical force (magnetic) to efficiently deliver siRNA to all retinal layers. Interestingly, the use of a cationic lipid formulation for siRNA transfection of mouse retinal cultures from the RPE side has been described [13], which seems surprising since lipid formulations form deposits at the bottom of the well, hindering diffusion through the retinal layers. Another study indicated that intravitreal application using the nonviral cationic polymer/lipid carrier Transit TKO led to the accumulation of siRNA, mainly restricted to the GCL, based on fluorescent-tagged siRNA localization [50]. Herein, we could not achieve efficient *VCP* silencing in retinal explants when siRNA was delivered by lipofection. Thus, our results stress the advantage of Magnetofection over lipofection and support that physical force is more likely to result in extensive transfection of all retinal layers. Our data show that efficient gene delivery depends on which side of the retina the siRNA is administered from. Using Reverse Magnetofection to approach the retina from the posterior RPE side, we obtained efficient siRNA delivery and silencing of VCP protein expression in all retinal layers. In contrast, when applying siRNA/MNPs complexes directly to the GCL in retinal explants (classical Magnetofection, Appendix A), gene silencing was less efficient compared to Reverse Magnetofection, especially in the INL and the GCL (data not shown), with a corresponding lower neuroprotection compared to the untreated control (Appendix A). The results are supported by others, who showed that plasmids introduced into bovine retinal explants on the GCL side, whether alone or complexed with cationic molecules, are not absorbed in the retinal cells behind the GCL [14]. Similarly, when applied from the GCL, electroporation is not efficient in transfecting retinal explants, but works when the explant is treated from the RPE side [40]. This is consistent with our findings and reinforces the evidence that a physical force (electric or magnetic) from the RPE side improves the efficacy of siRNA delivery and gene silencing throughout the retina. Indeed, viral vectors transfect all retinal layers efficiently, but with crucial inconveniences. Retroviruses are infectious only to mitotic cells, and thus retina precursor cells are the main targets that can be found exclusively in embryonic retinae [17]. Different serotypes of Adeno-associated virus (AAV) mediate cell subtype-specific transduction in mature explants [16]. However, only isolated patches across the explant can be infected with low infection efficiency (<4%). Additionally, viral vectors are not compatible with the delivery of synthetic siRNA. In this case, where genetic material is used to knockdown specific proteins, RNAi-mediated gene silencing relies on gene constructs containing appropriate promoters, whose activity is cell-type dependent and requires time-consuming evaluation [16,17,40]. Considering these limitations, as well as the safety issues of viral transfection [12], Reverse Magnetofection is the more suitable approach for gene delivery to the retina as it can efficiently deliver functional nucleic acid to all deeper retinal layers without side effects. Moreover, compared to other methods, Reverse Magnetofection is easy to handle. The entire protocol can be completed in just 1 h without the use of any device. The strong magnetic fields routinely used in magnetic resonance imaging—using MNPs as contrast agents—are safe [51]. MNPs in the eye are safe as well and biocompatible, as demonstrated in long-term toxicity studies in rat eyes with different particle sizes [46,52,53]. We confirmed this on retinal explants, as no signs of toxicity or immunogenicity were detected. *VCP* was downregulated in all layers of the retina, but more importantly, this also achieved a desired neuroprotective effect. This is consistent with our previous results, showing that inhibition of VCP function strongly attenuates retinal degeneration in flies [22] as well as in rodent models of RHO P23H [23]. Compared to drug therapy, RNAi could thus be an alternative bearing fewer side effects. Currently, VCP inhibitors can only dissolve in dimethyl sulfoxide (DMSO), from which neurotoxicity and retinal side effects are known [54]. On the other hand, siRNA is short-lived. Considering the long-term neuroprotective effect achieved with pharmacologic, small molecule-based application of VCP inhibitors [23], a longer follow-up of RNAi-based treatment of retinal tissue would be needed to validate whether long-term protection via anti-VCP RNAi could also be when delivered by Magnetofection.

In any case, the application of Reverse Magnetofection offers opportunities for potential therapeutic applications in disease models and eventually in humans. When potentially applied as a therapeutic in the future in vivo, VCP RNAi may be administered topically or via a contact lens placed on the eye, followed by subsequent magnetic guidance. Especially the latter holds for future in vivo investigations and bears potential for future clinical application that avoid the side-effects of invasive, intravitreal applications.

To conclude, our study provides a nontoxic, biocompatible, and robust strategy for delivery of therapeutic RNAi to retinal explants. Future in vivo application in disease models for adRP may show whether long term neuroprotection can be achieved within the intact eye. If so, this technology can contribute to the development of minimally invasive RNA-based therapies to treat various ocular disorders.

## 4. Material and Methods

### 4.1. Study Approval and Animals

All procedures were approved by the Tübingen University committee on animal protection (§4 registrations from 26.10.2018 and AK 15/18 M) and performed in compliance with the Association for Research in Vision and Ophthalmology ARVO Statement. Animals for in vitro studies were kept in the Tübingen Institute for Ophthalmic Research animal housing facility, under standard white cyclic lighting, had free access to food and water, and were used irrespective of sex unless stated otherwise. All efforts were made to minimize the number of animals used and their suffering. Homozygous P23H rhodopsin transgenic rats (produced by Chrysalis DNX Transgenic Sciences, Princeton, NJ, USA) of the SD-Tg(P23H)1Lav (P23H-1) line 1 were kindly provided by Dr. M. M. LaVail (University of California, San Francisco, CA, USA) or by the Rat Resource and Research Center (RRRC) at the University of Missouri. The rats were bred in the animal housing facility of the Institute for Ophthalmic Research in Tübingen. We used heterozygous RHO P23H rats obtained by crossing homozygous P23H rhodopsin transgenic rats with wild-type CD rats (CDH IGS Rat; Charles River, Germany) to reflect the genetic background of the adRP.

### 4.2. XPMag Synthesis and Characterization of VCP siRNA/XPMag Complex

XPMag (OZ Biosciences, Marseille, France) were synthesized by chemical coprecipitation of iron precursors in an alkaline environment and functionalized using a library of proprietary cationic lipids or polymers according to a procedure described elsewhere [55]. All siRNAs were nonmodified 21-mer with double-stranded RNA bases and double single-stranded RNA bases in 3′. VCP siRNA mouse (152438, Ambion, Thermo Fischer Scientific, Waltham, MA, USA) (Sense Sequence (SS):5′-GCGAUGCUUUAAUGAAAGATT-3′, Antisense Sequence (AS):3′-UCUUUCAUUAAAGCAUCGCCG-5′), validated VCP siRNA targeting human, mouse, and rat (s14765, Ambion, Thermo Fischer Scientific, Waltham, MA, USA) (SS:5′-GAAUAGAGUUGUUCGGAAUTT-3′, AS:3′-AUUCCGAACAACUCUAUUCAT-5′), and validated VCP siRNA targeting human (s14767, Ambion, Thermo Fischer Scientific, Waltham, MA, USA) (SS:5′-GGCUCGUGGAGGUAACAUUTT-3′, AS:3′-AAUGUUACCUCCACGAGCCTT-5′ were used in this study. The universal scrambled siRNA (which does not target any human, rat, or mouse genes) (OriGene, SR30004, Rockville, MD, USA) was used as a negative control. siRNA sequence was the following: SS:5′-CGUUAAUCGCGUAUAAUACGCGUAT-3′, AS:3′-AUACGCGUAUUAUACGCGAUUAACGAC-5′. For the VCP siRNA complexation kinetics of XPMag, the VCP siRNA/XPMag complexes formation was assessed by dynamic light scattering and electrophoretic light scattering measurements. Briefly, 100 µL of different concentrations of VCP siRNA prepared from 1 µM siRNA stock solution diluted with RNAse-free water were added to 2 µL of XPMag, which was previously placed in the wells of a sterile 96-well plate. Each condition was incubated at room temperature (RT) for 30 min. Then, complexes were diluted to 1mL with sterile MilliQ water in disposable cuvettes under sterile conditions and equilibrated at 25 °C for 1 min before analysis. Gently pipetting of the mixture ensured correct dispersion of the complexes. Importantly, the proportions between the volumes of siRNA, MNPs and MilliQ water were those used during Reverse Magnetofection of VCP siRNA in retinal explants. The average hydrodynamic size was recorded first and calculated from nine consecutive DLS measurements each of 30 s, followed by electrophoretic light scattering measurements to calculate the zeta potential from 30 consecutive measurements each of 10 s.

### 4.3. Transmission Electron Microscopy (TEM)

In order to perform TEM on magnetic beads composing XPMag, the formulation (nanoparticles + solvent) was sonicated for 30 min in an ultrasound bath (Bioblock Scientific; Strasbourg, France) before transferring 1–2 µL on holey carbon film covered copper grids. The solvent was left to dry at RT for 15 min before exposition to the electron beam accelerated at 200 kV. Magnetic nanoparticles composing XPMag were finally imaged using a JEOL 2100 analytical electron microscope (JEOL, Tokyo, Japan).

### 4.4. Cell Culture and Transfections

661W (immortalized cone photoreceptor cell line derived from the retinal tumor of a mouse expressing the SV40 T antigen) [56] and hTERT RPE-1 (human retinal pigmented epithelial cells expressing the human telomerase reverse transcriptase subunit) [57,58] cell lines were kindly provided by INSERM (UMR8U1112, Université de Strasbourg, Strasbourg, France). Cells were cultured in T75 plates at 37 °C, 5% CO_2_ in DMEM (Gibco, Thermo Fischer Scientific, Waltham, MA, USA) supplemented with 10% FBS (Corning Media Tech™, Corning, NY, USA) and 1% penicillin/streptomycin (Gibco, Thermo Fischer Scientific, Waltham, MA, USA).

### 4.5. Organotypic Culture of Explants

In vitro retina cultures were prepared according to published protocols [11,59]. Briefly, PN12 animals were sacrificed, the eyeballs were removed and collected in a serum-free R16 culture medium (Invitrogen Life Technologies, 07490743A, Leicestershire, UK). The eyes were enucleated and pretreated with 0.12% Proteinase K (MP Biomedicals, 0219350490, Illkirch-Graffenstaden, France) for 15 min at 37 °C in R16 medium. Retinas were dissected, cut into four fragments, and placed on culture inserts (0.4 μm Polycarbonate membrane, Corning Life Sciences, CLS3412, Corning, NY, USA) with ganglion cells up. The retinal explants were cultured in 1 mL serum-free culture medium consisting of Neurobasal A (Gibco, Thermo Fischer Scientific, 21103049, Waltham, MA, USA) supplemented with 2% B-27 supplement (Gibco, Thermo Fischer Scientific, 17504044, Waltham, MA, USA), 1% N_2_ supplement (Gibco, Thermo Fischer Scientific, 17502048, Waltham, MA, USA), 1% penicillin solution (Gibco, Thermo Fischer Scientific, 15140-122, Waltham, MA, USA), and 0.4% GlutaMax (Gibco, Thermo Fischer Scientific, 35050061, Waltham, MA, USA). Explants were maintained at 37 °C in a humidified atmosphere with 5% CO_2_.

### 4.6. Classical Magnetofection of VCP siRNA to Immortalized Mouse 661W Retinal and Human hTERT-RPE1 Cell Lines

siRNAs were transfected by Magnetofection according to the OZ Biosciences protocol. The siRNA concentrations described below correspond to the final volume in the well after the addition of the siRNA magnetic complexes. Complexes were prepared at RT for 30 min in serum-free DMEM. Mouse retinal 661W and human hTERT-RPE1 cells were seeded in 24-well plates at a concentration of 50,000 cells/well in 400 µL of their respective medium. The next day, complexes were prepared by adding 1 µM VCP siRNA or scrambled siRNA in 100 µL culture medium without any supplement to 2 µL of XPMag (OZ Biosciences, Marseille, France) for a final siRNA concentration of 25 nM in the well. Then, VCP siRNA/XPMag complexes were added directly to the cells and the culture plate was placed on a super-magnetic plate (OZ Biosciences, Marseille, France) to transfect cells through Magnetofection for 30 min at 37 °C and 5% CO_2_. The magnet was then removed, and the cells were incubated for a further 48 h for human hTERT-RPE1 and 72 h for 661W mouse retinal cells. For Lullaby (OZ Biosciences, Marseille, France) transfection, 25 nM VCP siRNA or 2 µL Lullaby were mixed with 50 µL serum-free DMEM. The siRNA solution was added to the Lullaby reagent and carefully mixed by pipetting up and down. The mixtures were incubated for 30 min and added to 400 µL culture medium. The cells were further incubated 48 h for human hTERT-RPE1 and 72 h for 661W mouse retinal cells. Each experiment included cells that did not receive any treatment as a control. *VCP* gene knock-down was qualitatively assessed by fluorescence microscopy (Eclipse TE2000, Nikon, Melville, NY, USA).

### 4.7. Classic-, Reverse Magnetofection and Lipofection of VCP siRNA in RHO P23H Rat Retinal Explants

For classical Magnetofection, 50 nM siRNAs of VCP and scrambled siRNA were complexed with 100 µL Neurobasal medium and complexed with 2.1 µL XPMag for 30 min. After complexation, 50 µL of the mixture was administered to the top of each retina on the vitreous side. Then, 1 mL of Neurobasal medium was added to a well of 6-well culture plates. The super-magnetic plate was placed under the 6-well plate of retinal cultures for 30 min to pull down the complexes through the retina. For Reverse Magnetofection, complexes of siRNA and MNP were prepared using 2.1 µL of XPMag mixed with 50 nM VCP siRNA or scrambled siRNA and incubated 30 min at RT in 100 µL of serum and complement free Neurobasal medium. Then, magnetic complexes were added to 900 µL of serum-free Neurobasal medium preplaced in 6-well culture plates. Then, nitrocellulose culture membrane supports, each one carrying a retina explant, were placed onto the medium containing the complexes. At this time, Reverse Magnetofection was applied by placing the super-magnetic plate above the lid of the culture plate. After 30 min of Reverse Magnetofection, the magnetic field was removed, and the explants were incubated under standard conditions. A total of 24 h later, the culture medium was replaced with fresh medium for another 48 h before *VCP* silencing evaluation was performed. Untreated retinae, retinae treated with scrambled siRNA, and retinae treated with XPMag that were not complexed with siRNA were used as controls. For Lullaby transfection, 50 nM VCP siRNA or 4 µL Lullaby was mixed with 50 µL Neurobasal medium. The siRNA solution was added to the Lullaby reagent and carefully mixed by gently pipetting up and down. The mixtures were incubated for 30 min and added to 900 µL culture medium. In all cultures, the culture medium was replaced with fresh culture medium after 24 h, and retinal cultures were incubated for additional 48 h prior to evaluation of *VCP* silencing.

### 4.8. Evaluation of VCP Silencing by Immunocytochemistry (ICC) in Retinal Cell Lines

For ICC experiments of VCP, 661W mouse retinal cells and human hTERT-RPE1 cells were cultured in a 24 well plate on 0.5% gelatin-coated glass coverslip. After siRNA transfection, the cells were washed once with Dulbecco’s phosphate-buffered saline (DPBS) and fixed in 4% paraformaldehyde for 10 min. Subsequently, the cells were washed twice with 1×DPBS and incubated for 1 h in blocking solution (0.1% Triton X-100, BSA 2% in 1×DPBS). After, immunofluorescence experiments were performed using a 1:100 dilution of mouse monoclonal anti-VCP (Appendix A) in blocking solution and a 1:200 dilution of appropriate anti-mouse IgG Alexa Fluor™ 568 dye-conjugated antibodies (Appendix A) in 1xDPBS. The nuclei were stained using DAPI (Roche, Switzerland) at 3.5 µM in 1×DPBS. Slides were mounted with Vectashield media for fluorescence microscopy (Vector laboratories, Peterborough, UK) and imaged under fluorescence microscopy (Nikon Eclipse 2000, Melville, NY, USA).

### 4.9. Evaluation of VCP Silencing by Western Blotting in Human hTERT-RPE1 Cell Line

For Western blot analyses, human hTERT-RPE1 cells were cultured in a 24-well plate. After siRNA transfection, the cells were washed once with DPBS and were lysed with ice-cold Radioimmunoprecipitation assay buffer (RIPA buffer: 50 mM Tris-HCl pH8, 150 mM NaCl, 1 mM EDTA, 1% NP-40, 0.1% SDS and 0.05% sodium deoxycholate) containing 2% (*v*/*v*) mammalian protease inhibitor cocktail (Sigma Aldrich, Munich, Germany). Lysates were rotated for 30 min at 4 °C and centrifuged for 15 min at 12,000× *g* at and 4 °C, diluted in sample loading solution. Proteins were then transferred onto nitrocellulose membranes. Membranes were blocked with 5% Marvel milk in PBS 0.1% Tween 20 for 1 h and probed with appropriate primary antibodies mouse monoclonal anti-VCP and anti-β-Actin (Appendix A) overnight at 4 °C. Proteins of interest were detected using anti-mouse IgG kappa binding protein-coupled to horseradish peroxidase. HRP signal was visualized with the ECL detection kit (Thermo Fischer Scientific, 32106, Waltham, MA, USA).

### 4.10. Preparation of Retinal Sections and Immunohistochemistry for Frozen Retinal Slides

The tissues were immersed in 4% paraformaldehyde in 0.1 M phosphate buffer (PB; pH 7.4) for 45 min at 4 °C, followed by cryoprotection in graded sucrose solutions (10%, 20%, 30%) and embedded in cryomatrix (Tissue-Tek^®^ OCT Compound, Sakura^®^ Finetek, VWR, 4583, Radnor, PA, USA). Radial sections (14 µm thick) were collected, air-dried, and stored at −20 °C. Retina sections were incubated overnight at 4 °C with the primary antibodies (Appendix A) diluted in blocking solution. Fluorescence immunocytochemistry was performed using either IgG Alexa Fluor™ 568 dye-conjugated goat anti-mouse IgG or Alexa Fluor™ 488 dye-conjugated goat anti-rabbit (Appendix A). Negative controls were carried out by omitting the primary antibody. DAPI (Vectashield Antifade Mounting Medium with DAPI; Vector Laboratories, H-1200, Peterborough, UK) was used as a nuclear counterstain. Finally, the slides were mounted with Fluoromount Aqueous Mounting Medium (Thermo Fisher Scientific, F4680, Waltham, MA, USA and imaged with the Axio Imager Z.1 ApoTome microscope equipped with a Zeiss Axiocam MRm digital camera (Jena, Germany).

### 4.11. TUNEL Assay

TUNEL assay [60] was performed using an in situ cell death detection kit conjugated with fluorescein isothiocyanate (Roche, 11684795910, Mannheim, Germany). DAPI (Vectashield Antifade Mounting Medium with DAPI; Vector Laboratories, H-1200, Peterborough, UK) was used as a nuclear counterstain.

### 4.12. Data Analyses

Results obtained from immunoblots, and histological measurements of the mean fluorescence intensity of VCP, RHO, TUNEL positive cell death, and the number of rows of photoreceptor’s nuclei in the ONL were analyzed using one way ANOVA by GraphPad Prism 7.05 for Windows followed by Tukey multiple comparisons test. *p* < 0.05 was considered significant.

## Figures and Tables

**Figure 1 pharmaceutics-13-00225-f001:**
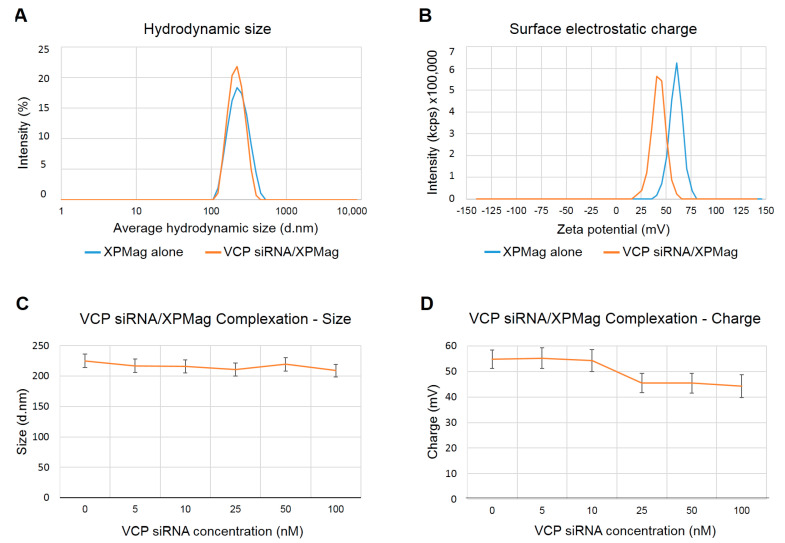
Characterization of valosin-containing protein (VCP) siRNA/XPMag complexation. (**A**) Hydrodynamic size distribution of XPMag alone (blue line) and VCP siRNA (25 nM)/XPMag complexes (orange line) measured by dynamic light scattering technique. The addition of VCP siRNA to XPMag did not influence the average hydrodynamic size of XPMag nanoparticles after 30 min of complexation. The graphic shows the distribution of the percentage of particle’s intensities versus average hydrodynamic size in diameter (d.nm). Average values are reported on a logarithmic scale. (**B**) Surface electrostatic charge of XPMag alone (blue line) the complexed with VCP siRNA (orange line), measured by electrophoretic dynamic light scattering. The surface charge of XPMag shifted towards a lower value upon the addition of VCP siRNA (orange line) without alteration of the peak shape. The graphic shows the scattering intensity recorded in photons per second (kcps) versus voltage-dependent electrophoretic mobility (mV). (**C**) Effect of VCP siRNA concentration on XPMag hydrodynamic size after 30 min of MNPs-siRNA complexation. Complexes obtained with concentrations of VCP siRNA from 5 to 100 nM did not differ in their hydrodynamic size, error bars are standard deviation, *n* = 9. (**D**) Effect of VCP siRNA on XPMag electrostatic surface after 30 min of MNPs-siRNA complexation. XPMag electrostatic surface decreased as a function of VCP siRNA concentration. Error bars correspond to standard deviation, *n* = 30. Abbreviations: d. nm: Diameter. nanometer; kcps: kilo counts per second, mV: millivolts.

**Figure 2 pharmaceutics-13-00225-f002:**
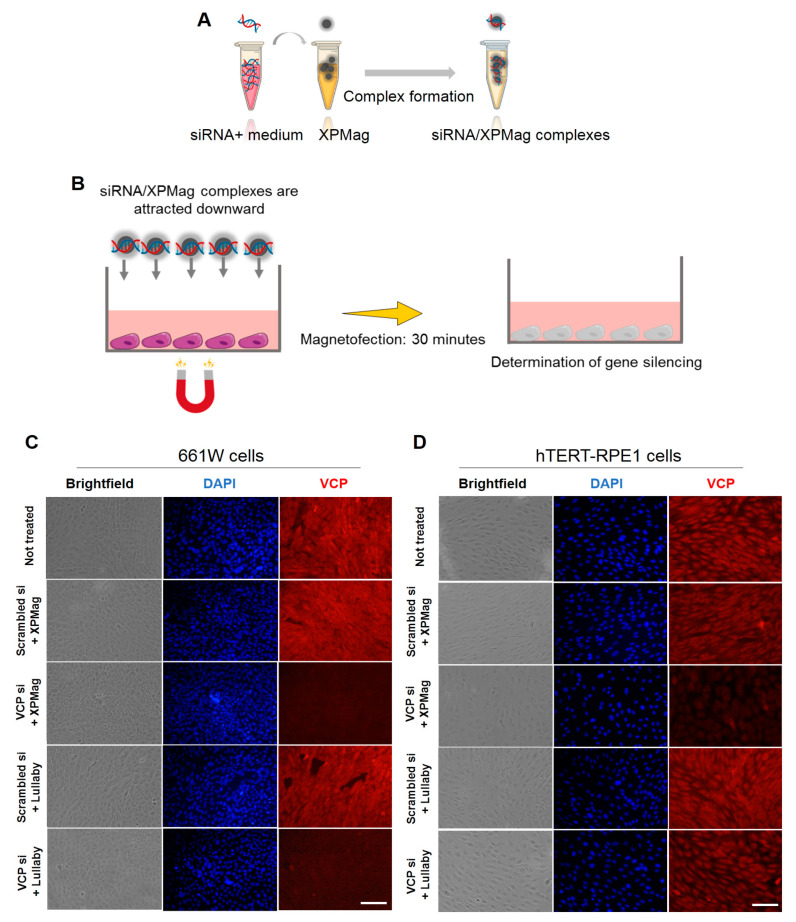
XPMag enhances in vitro *VCP* silencing in mouse retinal 661W and human hTERT-RPE1 cells by forming complexes with VCP siRNA via classic Magnetofection. (**A**,**B**) Illustration of the classic Magnetofection technique used for transfection of siRNA. First, siRNA is added to XPMag MNPs, followed by an incubation time of 30 min to allow the correct formation of MNPs/siRNA complexes (**A**), then the complexes are added to the medium of the cell culture model and then attracted towards the cells by placing a magnet under the culture plate during 30 min for classic Magnetofection (**B**). The siRNA transfection was assessed 48–72 h after treatment. Representative images of VCP expression (red) and cell nuclei with DAPI (blue). Classic Magnetofection using 2 μL XPMag and 25 nM VCP siRNA showed unaltered cell monolayer integrity and a considerable decrease in VCP signal intensity in treated mouse retinal 661W cells at 72 h (**C**) and human hTERT- RPE1 cells after 48 h (**D**). Lullaby complexed with VCP siRNA was used to compare the efficiency of lipofection with Magnetofection. Magnetofection of scrambled siRNA complexed with XPMag and XPMag alone were used as negative control groups. Scale bar 200 µm. Abbreviations: VCP si: VCP siRNA, Scr si: scrambled siRNA.

**Figure 3 pharmaceutics-13-00225-f003:**
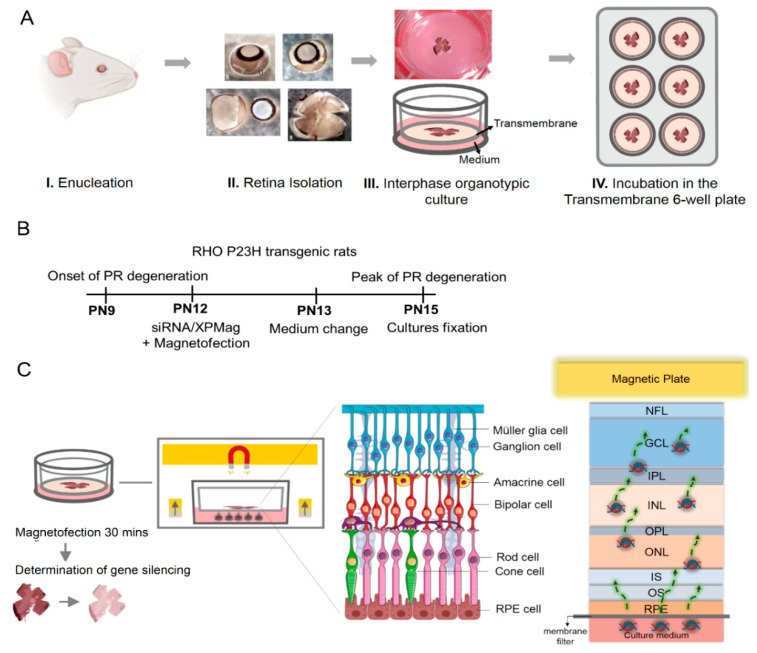
Schematic representation of the preparation of retinal organotypic cultures and Reverse Magnetofection technique used for transfection of siRNA. (**A**) Preparation technique for retina explants. The eyes were enucleated in an aseptic environment (I). Sclera, choroid, and lens were carefully removed, leaving only the neural retina and RPE. Four radial incisions were made to flatten the retina (II). The entire retina with its adherent RPE was mounted flat polycarbonate Transwell^®^ membranes, with the GCL facing up (III). Inserts carrying the retinae were transferred to 6-well plates and incubated for 3 days (IV). (**B**) Timeline of photoreceptor degeneration in RHO P23H heterozygous transgenic rats and experimental design. Photoreceptor degeneration in RHO P23H rats starts around postnatal (PN) 9 and reaches its peak at PN15. Accordingly, the RHO P23H retinae were isolated from PN12 and silenced for 3 days (72 h) until the peak of degeneration at PN15. (**C**) Illustration of Reverse Magnetofection in retinal organotypic cultures. The Reverse Magnetofection is achieved by adding the siRNA/XPMag MNPs complexes to the culture medium, followed by placing the magnet over the culture plate to attract the complexes to the tissue up to the GCL. The effect of the magnet on the siRNA/MNPs complexes allows siRNA delivery into deeper layers of the explant that are not directly exposed to the culture medium. Abbreviations: PR: photoreceptor, NFL: nerve fiber layer, GCL: ganglion cell layer, IPL: inner plexiform layer, INL: inner nuclear layer, OPL: outer plexiform layer, ONL: outer nuclear layer, IS: inner segment, OS: outer segment and RPE: retinal pigmented epithelium. Schematic representation of the retinal layers was prepared using Smart Servier software. The illustration of the magnetic flow on the different retinal layers was inspired and modified from Bassetto et al. [24].

**Figure 4 pharmaceutics-13-00225-f004:**
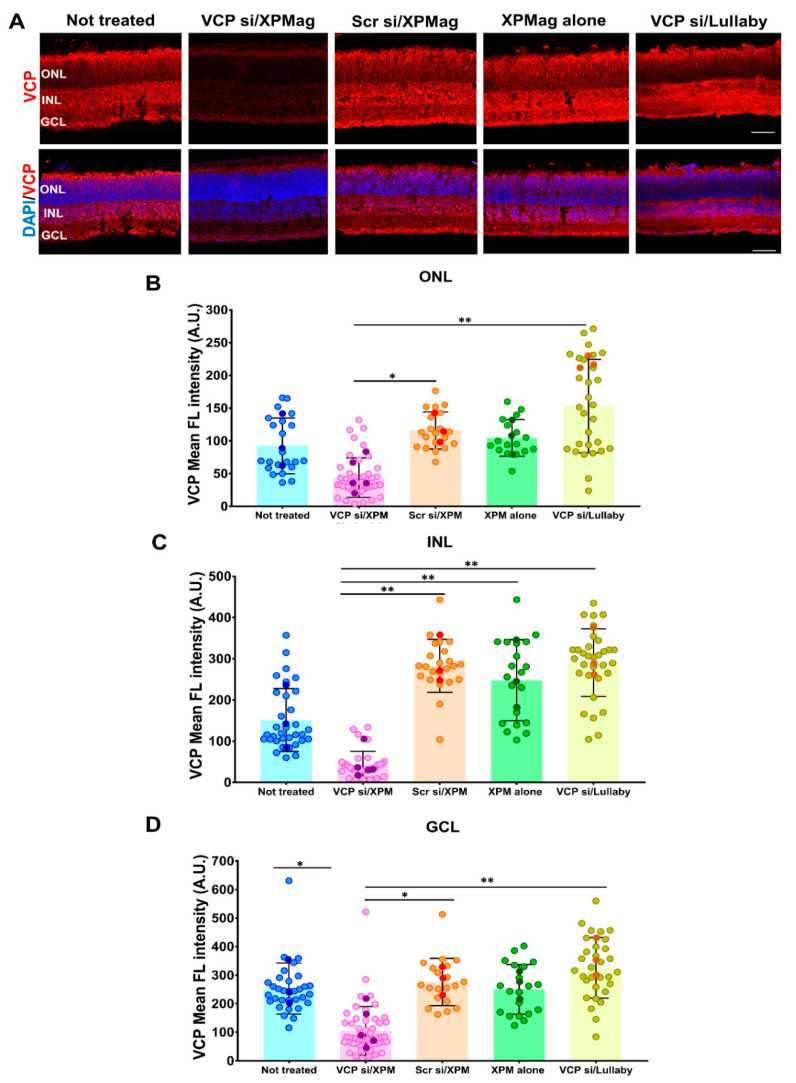
Efficient gene silencing of *VCP* in RHO P23H rat retinal organotypic culture by Reverse Magnetofection through the entire layers of the retinal explant at 72 h after treatment. Retinae of RHO P23H transgenic rats were explanted on the postnatal day 12 and cultured for three days. Retinae were treated with 50 nM VCP si/XPMag, 50 nM Scr si/XPMag, and XPMag alone by Reverse Magnetofection and VCP si/Lullaby by lipofection. (**A**) Explants were stained with specific VCP antibody (red) using nuclei counterstaining with DAPI (blue). (**B**–**D**) Quantification of VCP immunofluorescence intensity in the ONL, INL and GCL. For each layer, a complete region of each image was selected, and the mean fluorescence intensity was assessed using the Zen 3.2 software. Bar graph shows the mean fluorescent intensity in (**B**) ONL, (**C**) INL, and (**D**) GCL upon Reverse Magnetofection of VCP siRNA/XPMag as compared to control conditions. A significant decrease in VCP signal intensity in all three different layers after Reverse Magnetofection of VCP siRNA/XPMag compared to the other groups. Scale bar 50 μm. The values were quantified by evaluating multiple images (circles) from three retinae (*n* = 3) per treatment for not treated, Scr si/XPMag, XPMag alone, and VCP si/Lullaby groups, five retinae (*n* = 5) per treatment for VCP si/XPMag group. Data are presented as mean ± SD, and one-way ANOVA analysis was performed at * *p* < 0.05 and ** *p* < 0.01. The error bars correspond to the standard deviation. Abbreviations: VCP si: VCP siRNA, Scr si: scrambled siRNA, XPM: XPMag, FL: fluorescence, ONL: outer nuclear layer, INL: inner nuclear layer, and GCL: ganglion cell layer.

**Figure 5 pharmaceutics-13-00225-f005:**
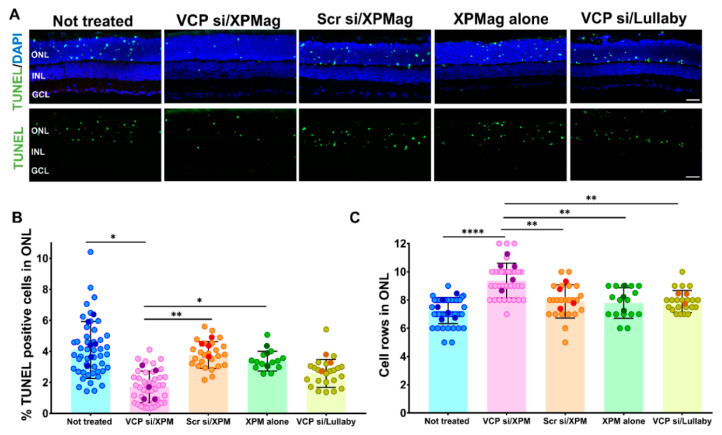
Reverse Magnetofection does not induce cellular toxicity and *VCP* silencing is neuroprotective in RHO P23H retinal explants. Retinae of RHO P23H transgenic rats were explanted on the postnatal day 12 and cultured in vitro for three days. Retinae were treated with 50 nM VCP si/XPMag, 50 nM Scr si/XPMag, and XPMag alone by Reverse Magnetofection and VCP si/Lullaby by lipofection. (**A**) The explants were stained with the TUNEL assay to differentiate photoreceptors undergoing cell death (green) using nuclei counterstaining with DAPI (blue). (**B**) Bar graph shows the percentage of TUNEL-positive cells in the ONL. After treatment with VCP siRNA/XPMag via Reverse Magnetofection, a significant decrease in the percentage of dying cells was observed compared to the other groups. (**C**) Quantification of the number of photoreceptor cell rows in ONL, calculated by counting DAPI-stained cells in a linear row in three different vertical sections. Retinae treated with VCP siRNA/XPMag via Reverse Magnetofection showed significant preservation of photoreceptor cell rows. Scale bar 50 μm. The values were quantified by scoring several images (circles) from six retinae (*n* = 6) per treatment for not-treated, five retinae (*n* = 5) per treatment for VCP si/XPMag, four retinae (*n* = 4) per treatment for Scr si/XPMag, and three retinae (*n* = 3) per treatment for XPMag alone and VCP si/Lullaby groups. The data are presented as mean ± SD, and one-way ANOVA analysis was performed at * *p* < 0.05, ** *p* < 0.01, **** *p* < 0.0001. The error bars correspond to the standard deviation. Abbreviations: VCP si: VCP siRNA, Scr si: scrambled siRNA, XPM: XPMag, FL: fluorescence, ONL: outer nuclear layer, INL: inner nuclear layer, and GCL: ganglion cell layer.

**Figure 6 pharmaceutics-13-00225-f006:**
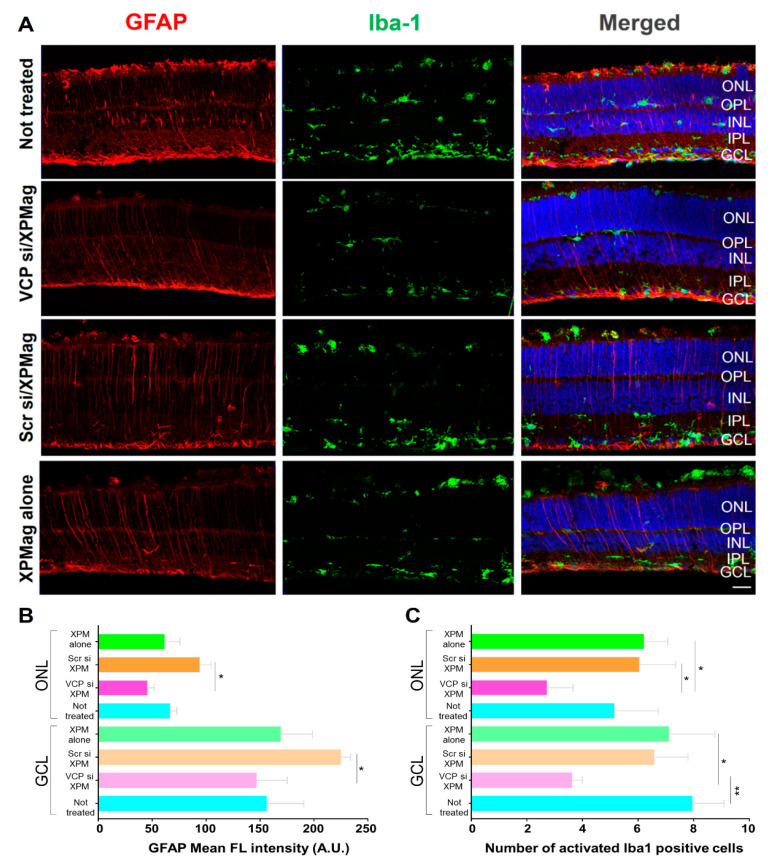
Reverse Magnetofection does not activate associated gliosis or microglial activation and reducing *VCP* expression decreased retinal stress and inflammation in RHO P23H retinal explants. Retinae of RHO P23H transgenic rats were explanted on postnatal day 12 and cultured in vitro for three days. Retinae were treated with 50 nM VCP si/XPMag, 50 nM Scr si/XPMag, and XPMag alone using Reverse Magnetofection. (**A**) Immunofluorescence labeling in cryosections designates the location of glial fibrillary acidic protein (GFAP) (red staining) and ionized calcium-binding adapter molecule 1 (Iba1) (green staining) as counterstaining of the nuclei by DAPI (blue). (**B**) Quantification of GFAP mean immunofluorescence intensity in the ONL and GCL. ONL or GCL were selected for each image, and the mean maximum intensity was assessed using the Zen 2.3 software. Reverse Magnetofection does not further activate retinal gliosis, and *VCP* silencing reduced the GFAP fluorescence intensity in the ONL and GCL compared to the scrambled sıRNA treated retinae. (**C**) Quantification of activated (round-shaped) Iba1 positive cells in ONL and GCL. Retinae treated with VCP siRNA/XPMag via Reverse Magnetofection showed a significant decrease in retinal inflammation. The values were quantified by scoring several images from three retinae (*n* = 3) per treatment for not treated, VCP si/XPM, Scr/XPM, and XPM alone. Data plotted as mean ± SD. One-way ANOVA, * *p* < 0.05 and ** *p* < 0.01. Scale bar 50 μm. Abbreviations: VCP si: VCP siRNA, Scr si: scrambled siRNA, ONL: outer nuclear layer, OPL: outer plexiform layer, INL: inner nuclear layer, IPL: inner plexiform layer, GCL: ganglion cell layer, GFAP: glial fibrillary acidic protein, and Iba-1: ionized calcium-binding adapter molecule 1.

**Figure 7 pharmaceutics-13-00225-f007:**
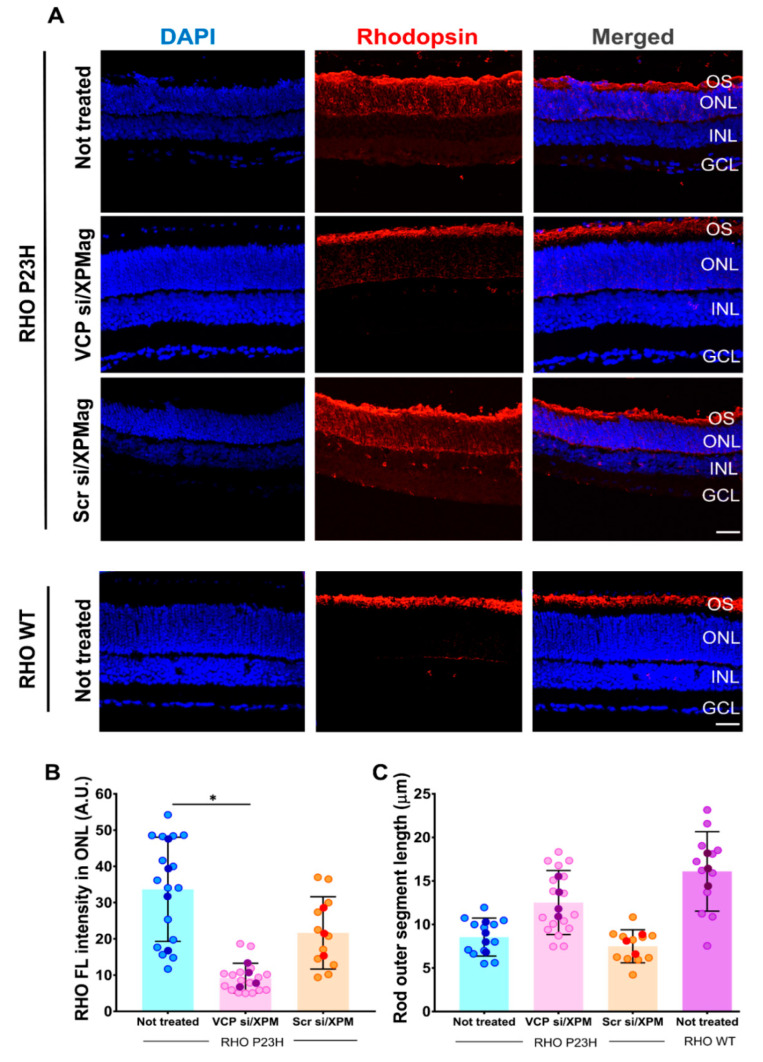
*VCP* downregulation via Reverse Magnetofection corrects the subcellular localization of rhodopsin (RHO) and increases the length of the OS in RHO P23H rat retina explants. Retinae of RHO P23H transgenic rats and WT rats were explanted on postnatal day 12, cultured in vitro for three days. RHO P23H retinae were treated with 50 nM VCP si/XPMag and 50 nM Scr si/XPMag by Reverse Magnetofection. The untreated age-matched WT retina group was used as a positive control. (**A**) The fluorescent marker in cryosections indicates the localization of RHO (red staining). (**B**) Quantification of RHO mean immunofluorescence intensity in the ONL. A central region of each image was selected, and the mean maximum intensity was assessed using the Zen 2.3 software. VCP silencing reduced the RHO fluorescence intensity in the ONL. (**C**) The mean length of the OS was higher in *VCP* silenced retinae cultures compared to untreated and scrambled siRNA controls (**C**). Scale bar 50 μm. The values were quantified by scoring several images (circles) from four retinae (*n* = 4) per treatment for not-treated RHO P23H, four retinae (*n* = 5) per treatment for VCP si/XPMag in RHO P23H, three retinae (*n* = 3) per treatment for Scr si/XPMag in RHO P23H, and three retinae (*n* = 3) per treatment for not-treated WT. The data are presented as mean ± SD, and one-way ANOVA analysis was performed at * *p* < 0.05. The error bars correspond to the standard deviation. Abbreviations: VCP si: VCP siRNA, Scr si: scrambled siRNA, OS: outer segment, ONL: outer nuclear layer, INL: inner nuclear layer, and GCL: ganglion cell layer, RHO: rhodopsin.

## Data Availability

The data presented in this study are available in this article: Efficient Ocular Delivery of VCP siRNA via Reverse Magnetofection in RHO P23H Rodent Retina Explants.

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
