# Peer review of "Efficient Ocular Delivery of VCP siRNA via Reverse Magnetofection in RHO P23H Rodent Retina Explants"

_pharmaceutics, 2021, doi:10.3390/pharmaceutics13020225_

Round 1

Reviewer 1 Report

valosin-containing protein is involved in a number of disease states.  Therefore, the significance of the work is high.  Specific issues are,

  1. Particle size is large, what is the level of uptake?
  2. Not clear about the level of magnetofection.  What was the tissue damage at different levels?
  3. There is no positive control, such as VCP inhibitor small molecule used in the study
  4. Immune response parameters are not clear.

A major revision would be advised.

Reviewer 2 Report

In this manuscript, the authors investigated the reverse magnetofection approach for delivering siRNA/magnetic nanoparticle complexes into retinal explants, targeting VCP as a potential therapeutic target for adRP. The studies are well-designed, and the results would extend the future use of magnetofection for clinical applications. The presented data are significant and novel, but additional experiments are required to strengthen the authors’ conclusions.  
  1. In Fig. 1, the DLS analysis of the magnetic nanoparticles showed a single bell-shaped size distribution before and after siRNA complexation, and the negative shift of the zeta potential by siRNA. Since the DLS analysis is a mass measurement, it would be better to validate the sizes of individual magnetic nanoparticles with or without siRNA complexation by electron microscopy.
  2. In Fig. 2D, the authors presented that magnetofection-based VCP silencing in mouse retinal 661W and human hTERT-RPE1 cells. In the text, the authors described that this was assessed by Western blotting, although the data were not shown. This data should be presented with a quantitative analysis of the result. Otherwise, readers cannot understand how the silencing efficiency was.
  3. In Fig. 3 and 4, magnetic force was applied for 30 min to deliver siRNA/magnetic nanoparticles to the tissue explants. During this relatively short period, I wonder how the magnetic particles can distribute in the 3D tissue structure. Fig. 4A shows overall suppression of the VCP expression, but it would be more informative if the distribution of siRNAs in the tissue after the magnetofection.
  4. In Fig. 6, data quantification should be presented to show the potential therapeutic effect of the reverse magnetofection-based VCP silencing.
  5. In the discussion, it was described as "Physical methods, Physical methods, such as electroporation, successfully reach the retina from the RPE side; nevertheless, they are only successful when the retina is still immature [40]". Similarly, it was described as "Since viral vectors can transfect all retinal layers efficiently, but only in the early postnatal days [16,17]". Those descriptions could give an inaccurate impression to readers. It should be described more specifically. For example, all of the cells in adult retinal layers are resistant to electroporation and various viral victors, particular methods, or target cells specifically?

Reviewer 3 Report

In the manuscript entitled "Efficient ocular delivery of VCP siRNA via Reverse Magnetofection in RHO P23H rodent retina explants", the authors tried commercial magnetotransfection reagents in delivering siRNA to the retina. The data presented in the manuscript proved satisfied transfection efficiency and safty. Although novelty has some shortcomings, this paper provides sufficient data, which can provide reference for other studies. Therefore, it is recommended to accept.

Author Response

The authors want to thank the reviewer for the positive evaluation.

Round 2

Reviewer 1 Report

None

Reviewer 2 Report

The authors studied the reverse magnetofection for the delivery of siRNA targeting VCP as a potential therapeutic target for adRP in retinal explants. The concerns in the previous review were appropriately addressed and the revised manuscript reads well.